# Legacies of domestication, Neolithic diffusion and trade between Indian subcontinent and Island Southeast Asia shape maternal genetic diversity of Andaman cattle

**Arun Kumar De**[1]*, **Sneha Sawhney**[1], **Ramachandran Muthiyan**[1], **Debasis Bhattacharya**[1],
**Perumal Ponraj**[1], **Dhruba Malakar**[2], **Jai Sunder**[1], **T. Sujatha**[1], **Ashish Kumar**[3],
**Samiran Mondal**[4], **Asit Kumar Bera**[5], **P. A. Bala**[1], **Eaknath B. Chakurkar**[1]

1 Animal Science Division, ICAR-Central Island Agricultural Research Institute, Port Blair, Andaman and
Nicobar Islands, India, 2 Animal Biotechnology Centre, National Dairy Research Institute, Karnal, Haryana,
India, 3 CTARA, IIT Bombay, Mumbai, Maharashtra, India, 4 Department of Veterinary Pathology, West
Bengal University of Animal and Fishery Sciences, Kolkata, West Bengal, India, 5 Reservoir and Wetland
Fisheries Division, ICAR-Central Inland Fishery Research Institute, Barrackpore, Kolkata, West Bengal, India

* biotech.cari@gmail.com

University, INDIA

**Data Availability Statement:** All relevant data are
within the paper and its Supporting Information
files.

## Abstract

Andaman cattle is a precious indigenous livestock species endemic to Andaman and Nico-
bar Islands, India. Till date, origin and genetic makeup of the breed which is warranted for
breed conservation is not known. Moreover, the spread of zebu cattle from Indus valley to
different parts of Island Southeast Asia (ISEA) is not properly understood. Here, we report
the genetic diversity, population structure of Andaman cattle and their evolution in the con-
text of epicentre of zebu domestication and ISEA. High genetic diversity in complete mito-
chondrial D-loop sequences indicated the ability of the breed to withstand impending
climate change. Total 81 haplotypes were detected and all of them except three belonged to
*Bos indicus*. The presence of taurine haplotypes in Andaman cattle indicate introgression by
European-derived cattle. A poor phylogenetic signal of Andaman cattle with genetic affinities
with cattle of Indian subcontinent and ISEA was observed. The poor phylogenetic structure
may be due to multidirectional gene flow from Indian subcontinent and ISEA, with which
Andaman shares a close cultural and trade relationship from Neolithic age. We hypothesize
that Andaman cattle is the outcome of Neolithic diffusion from centre of zebu domestication
along with multidirectional commercial exchange between Indian subcontinent and ISEA.

## Introduction

The switch from hunting and gathering to agriculture during Neolithic age, popularly known
as 'Neolithic Revolution', is a landmark event in the history of human civilization [1, 2]. Dur-
ing this period, domestication of animals from their wild progenitors took place at different
locations across the globe [3–5]. Three proposed pathways of domestication exist; commensal
pathway, prey pathway and directed pathway [6–8] and cattle are believed to be domesticated

**Funding:** This work was supported by a Grant from Indian Council of Agricultural Research, New Delhi, India (Grant No. HORTCIARISIL201700800181). There was no additional external funding received for this study. The funder had no role in study design, data collection and analysis, decision to publish, or preparation of the manuscript.

**Competing interests:** The authors have declared that no competing interests exist.

through prey pathway [7], in which animals were domesticated to increase the predictability of a prey animal. Wild aurochs (*Bos primigenius*) is considered as the ancestor of modern cattle [9, 10]. The domestication process of cattle is not fully understood and several theories on this exist. Currently, there is a broad consensus that modern cattle were domesticated in two independent domestication events; zebu cattle (*Bos indicus*) were domesticated in Indus valley ca. 8000–7500 before present (B.P), whereas domestication of taurine cattle (*Bos taurus*) occurred in Fertile Crescent ca 10,500–10,000 B.P [11–16]. Based on coalescent simulation and Bayesian computation of mitochondrial DNA (mtDNA) sequences from domestic cattle from Near East dating from Neolithic age to Iron age, Bollongino *et al*. [17] estimated the founding population of taurine cattle as around 80 female aurochs. Beja-Pereira *et al*. [18] argued against the single origin of European cattle and postulated 'multiple origin hypothesis' for European cattle which suggested gene introgression from local aurochs and African cattle transported to Europe through maritime route. Of recent, Larson and Burger [7] proposed "Independent domestication and subsequent introgression' hypothesis, in which it was claimed that taurine cattle were domesticated and zebu cattle resulted from introgression of wild cattle into taurine cattle which were transported earlier. The domestication and spread of taurine cattle have been studied extensively [18–21] whereas little attention has been paid to study the expansion of zebu cattle lineage from its domestication centre to different parts of Indian subcontinent and Southeast Asia.

Andaman and Nicobar Islands, an archipelago in the Bay of Bengal, is a transitional biogeographic zone between South and Southeast Asia. Archaeological data indicates waves of human migration from Indian subcontinent to Southeast Asian countries in Neolithic age and cattle accompanied human migrations [22]. Moreover, historical indentures suggest that trade and cultural exchanges between Indian subcontinent and Southeast Asian countries were established around 290 BC through maritime routes [22] and maritime trades were mostly via Andaman and Nicobar Islands, an ideal natural harbour in the sea route between Indian subcontinent and Southeast Asia [23, 24]. Livestock including cattle were a major component of trade. Therefore evolutionary relationship of zebu cattle spread across Andaman and Nicobar islands, Indian subcontinent and Island Southeast Asia merits in-depth study.

Erosion of indigenous livestock genetic resources is felt as a major threat to biodiversity [25]. Many native breeds have good production potential with adaptation to local microenvironment. Specific indigenous breeds due to adaptation to different environmental challenges acquire unique allele combinations to thrive in a particular environment [26]. Therefore, conservation of livestock genetic resources is key to maintain world biodiversity [27, 28]. Lack of data on genetic characterization of breeds is a bottleneck in breed conservation programme. Andaman cattle (ANC) is an indigenous cattle germplasm of Andaman and Nicobar Islands. They are humped cattle with moderate body size and body colour are white, black, red or a mixture of all these. Udder and teats are small to moderate in size. Phenotypically, these cattle have resemblance with Red Sindhi, Sahiwal and Hariana cattle breeds of India [29]. Andaman cattle, notable for their tolerance to harsh climatic condition, are threatened by introgression of exotic germplasm introduced for crossbreeding purpose. A massive declining tread in the population of this breed has been observed over the last decade; the population declined from 29.51 thousand in 2012 to 20.92 thousand in 2019 [30] which reinforces the major conservation concerns for this breed. No information on genetic structure, diversity and phylogeography of Andaman cattle is available. Information on genetic structure, phylogeography and population genetic parameters are prerequisite for decision making regarding conservation of the breed [31, 32]. The present study aims to unveil the population genetic structure and genetic diversity of Andaman cattle and their evolutionary relationship with indigenous cattle distributed across Indian subcontinent and Island Southeast Asia (ISEA).

## Materials and methods

### Ethics approval

Ethical permission to carry out the work was granted by the Institute Animal Ethical Committee (IAEC) of ICAR-Central Island Agricultural Research Institute, Port Blair, Andaman and Nicobar Islands, India (Approval letter, ICAR-CIARI/IAEC/ASD/HORTI234/2456 dated 2[nd] December, 2020). Relevant standard national guidelines and regulations were followed throughout the study.

### Sample collection

Blood samples from 150 Andaman cattle were collected from native breeding tract spread across different parts of Andaman and Nicobar Islands [South Andaman (n = 50), North and Middle Andaman (n = 80) and Nicobar (n = 20)]. Information on sampling sites is presented in S1 Table. Andaman cattle breed was identified by careful examination of the distinct breed characteristics. Randomly chosen samples (2–3 samples/village) from genetically unrelated adult animals based on pedigree history and detailed interviews with owners were considered [33]. Before collection of samples, the animal owners were briefed about the purpose of the study and written consents were obtained from them. Approximately, 5 ml of blood sample was drawn from each animal by jugular venipuncture into a vacutainer containing EDTA. Samples were transported to laboratory maintaining cold chain.

### DNA extraction, PCR amplification and sequencing of mtDNA D-loop

Genomic DNA was isolated using a commercial kit (GSure Blood DNA Mini Kit, GCC Biotech India Pvt. Ltd, Kolkata, India, Cat. No. G4626), following the manufacturer's instructions. The quality and concentration of isolated DNA samples were checked using a BioSpectrometer (Eppendorf, Hamburg, Germany) and DNA samples were then stored at -20˚C until further use. Complete mitochondrial D-loop was amplified using the primers and PCR conditions described earlier by Yang *et al.* [34]. Amplified products were purified by using MinElute PCR Purification Kit, Qiagen, Cat. No. 28004 (New Delhi, India) and sequenced in both directions by Sanger dideoxy fingerprinting. The generated sequences were edited using Codon code Aligner v 9.0.1 (CodonCode Corporation, www.codoncode.com).

### Bioinformatics analysis

Alignment of the mitochondrial D-loop sequences was done by ClustalW [35]. implemented in MEGAX [36]. Nucleotide composition, number of transition and transversion, transition/transversion rate ratios and the overall transition/transversion bias were calculated in MEGAX [36]. Skewness of the sequences were calculated manually [37] based on the following formulas; AT Skew = (A-T)/(A+T) and GC Skew = (G-T)/(G+T). Population diversity and polymorphism parameters of Andaman cattle like number of polymorphic sites, nucleotide diversity, haplotype number and diversity were estimated in DnaSp v 6 [38]. For haplogroup assignment, we used standard sequences of cattle mtDNA haplogroups T1 (LC013968), T2 (AB117049), T3 (V00654), T4 (LC013966), I1 (AB268579, L27722), I2 (AB268559, EU177870), P (DQ124389), R (HQ184045) and Q (EU177867). To unravel the relationship of Andaman cattle with indigenous cattle population of Indian subcontinent and Island Southeast Asian countries, representative D-loop sequences of indigenous cattle from different countries representing Indian subcontinent and Island Southeast Asia were retrieved from GenBank (*www. ncbi.nlm.nih.gov*) and a summary of the information has been depicted in S2 Table. India was

classified into four geographical regions; Central India, Eastern India, Southern India and Western India. Phylogenetic tree was built using the Neighbor-Joining method [39] with the Tamura-Nei distance [40] implemented in MEGAX following 1,000 bootstrap replications. Bayesian phylogenetic analysis of the sequences was established using MCMC model in BEAST v1.10.4 [41]. Alignment gaps were excluded from the analysis. Evolutionary relationship among different haplotypes were established by median-joining network and minimum spanning network constructed in PopART ver. 1.7 [42]. Population differentiation was calculated by Wright's F-statistics [43] and assessment of genetic variance among and within the population was done by analyses of molecular variance (AMOVA) [44] in Arlequin v 3.5 [45] with 16,000 permutations. To delineate population expansion, mismatch distribution was calculated in DnaSp v 6 [38] and neutrality tests (Tajima's D test, Fu's FS test, Fu and Li's D test, Fu and Li's F test) were done in DnaSp v 6 [38] and Arlequin v 3.5 [45].

## Results

### Polymorphisms and haplotype diversity of Andaman cattle

The complete mtDNA D-loop sequence information of 150 Andaman cattle was generated and the generated sequence information was deposited to GenBank (https://www.ncbi.nlm.nih.gov/genbank/) with accession numbers MK872811-MK872960.

The average (mean ± SD) frequency of A, T, G and C was 33.19 ± 0.1637%, 27.85 ± 0.1304%, 13.43 ± 0.2059% and 25.53 ± 0.1499% respectively indicating a bias towards A+T nucleotides (61.04 ± 0.1872%) than G+C nucleotides (38.96 ± 0.1872%). AT skew and GC skew of the sequences were 0.08739 ± 0.003710 and -0.3494 ± 0.007432 respectively. A total of 51 transitions and 15 transversions were observed. The transition/transversion rate ratios were k1 = 7.415 (purines) and k2 = 6.367 (pyrimidines). The overall transition/transversion bias (R) was 3.157. DNA polymorphism analysis of the sequences detected 64 polymorphic or variable sites including 63 parsimony informative sites and one singleton variable site. The average number of nucleotide differences (k) was 8.748 and nucleotide diversity ($\pi$ ± SD) of the sequences was found to be 0.00952 ± 0.00076.

A total of 81 haplotypes (ANCHT1-ANCHT81) were identified in Andaman cattle with haplotype diversity (Hd ± SD) of 0.968 ± 0.008. Frequency distribution of the detected haplotypes is presented in S1 Fig. The predominant haplotypes detected were ANCHT24 (n = 19, Frequency = 12.666), ANCHT1 (n = 16, Frequency = 10.66) and ANCHT22 (n = 8, Frequency = 5.333). Seven haplotypes were represented by three sequences each, 15 haplotypes were represented by two sequences each and 56 haplotypes were represented by one sequence each. The haplotype map of the detected haplotypes is presented in Fig 1. Three major clusters were observed; 3 haplotypes (ANCHT9, ANCHT15 and ANCHT62) formed one cluster, 20 haplotypes (ANCHT1, ANCHT4, ANCHT6, ANCHT7, ANCHT11, ANCHT12, ANCHT16, ANCHT18, ANCHT19, ANCHT21, ANCHT39, ANCHT44, ANCHT50, ANCHT54, ANCHT55, ANCHT60, ANCHT67, ANCHT74, ANCHT77 and ANCHT78) formed another cluster and rest of the haplotypes formed the third cluster.

When the haplotypes of ANC were aligned with *Bos indicus* reference sequence (NC_005971) [46], 64 variable sites including 7 singleton variable sites and 57 parsimony informative sites were detected. The multiple sequence alignment (MSA) of the variable sites was depicted Fig 2. When the haplotypes of ANC were aligned with *Bos taurus* reference sequence (V00654) [47], 73 variable sites (13 singleton variable sites and 60 parsimony informative sites) were detected. The multiple sequence alignment of the variable sites was depicted in S2 Fig.

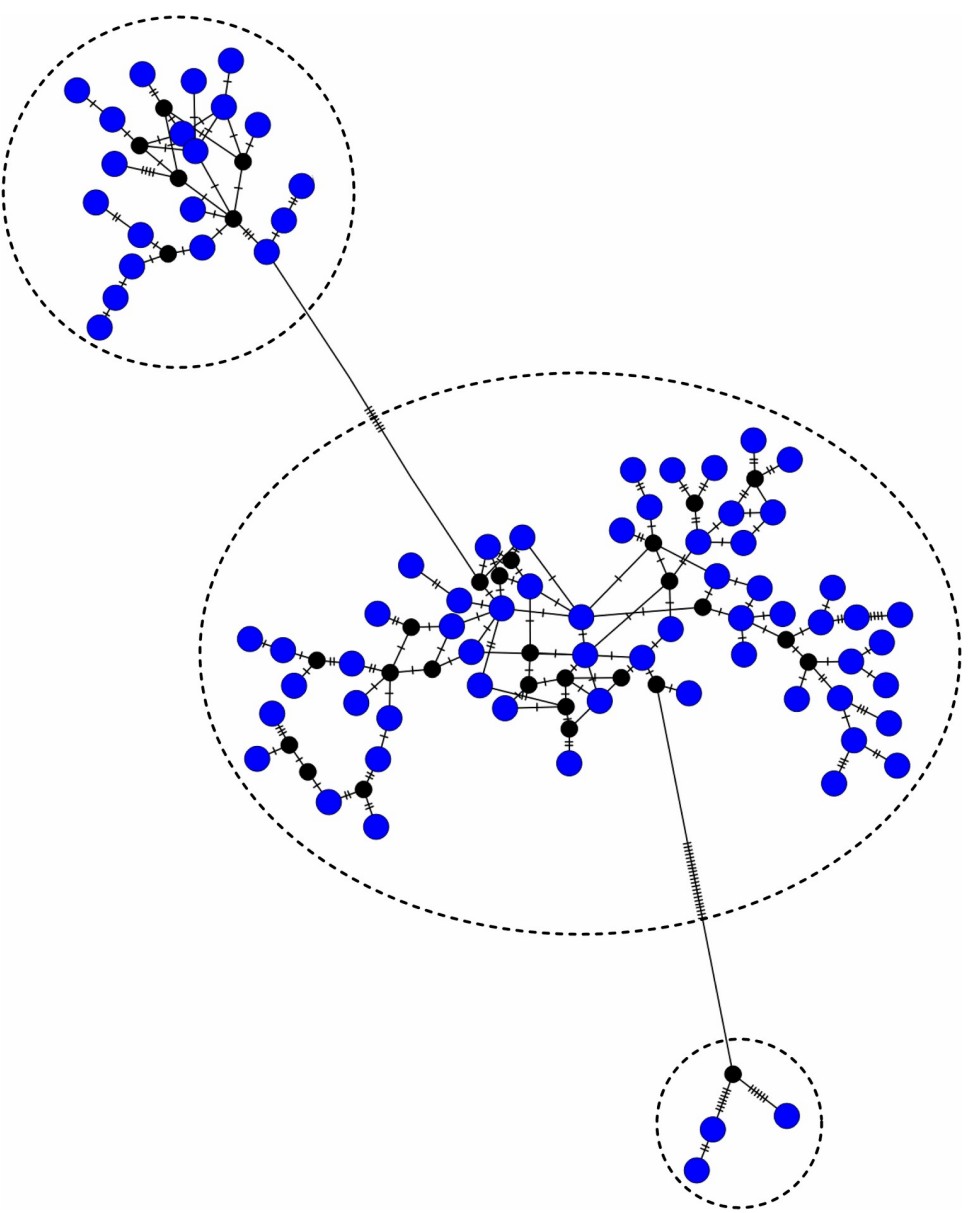

**Fig 1. Haplotype map of the detected haplotypes of Andaman cattle.** Dashed lines represent mutations. Network was drawn in PopART ver. 1.7 [42].

## Haplogroup assignment of the haplotypes of Andaman cattle

Based on the Neighbor Joining (NJ) phylogenetic tree of ANC haplotypes with standard cattle haplogroups (T1, T2, T3, T4, I1, I2, P, R, Q), all the haplotypes except ANCHT9, ANCHT15 and ANCHT62 belonged to *Bos indicus* I haplogroup (Fig 3 and S3 Table). The phylogenetic tree was constructed using 411 bp sequence information of hypervariable region of D-loop [48]. Among I haplogroup, 20 haplotypes belonged to haplogroup I2 and 58 haplotypes belonged to haplogroup I1 (Fig 3). The three haplotypes (ANCHT9, ANCHT15 and ANCHT62) belonged to haplogroup T/Q. It was found that 68% animals were under I1 haplogroup, 29.34% animals were under I2 haplogroup and 2.66% animals were under T/Q

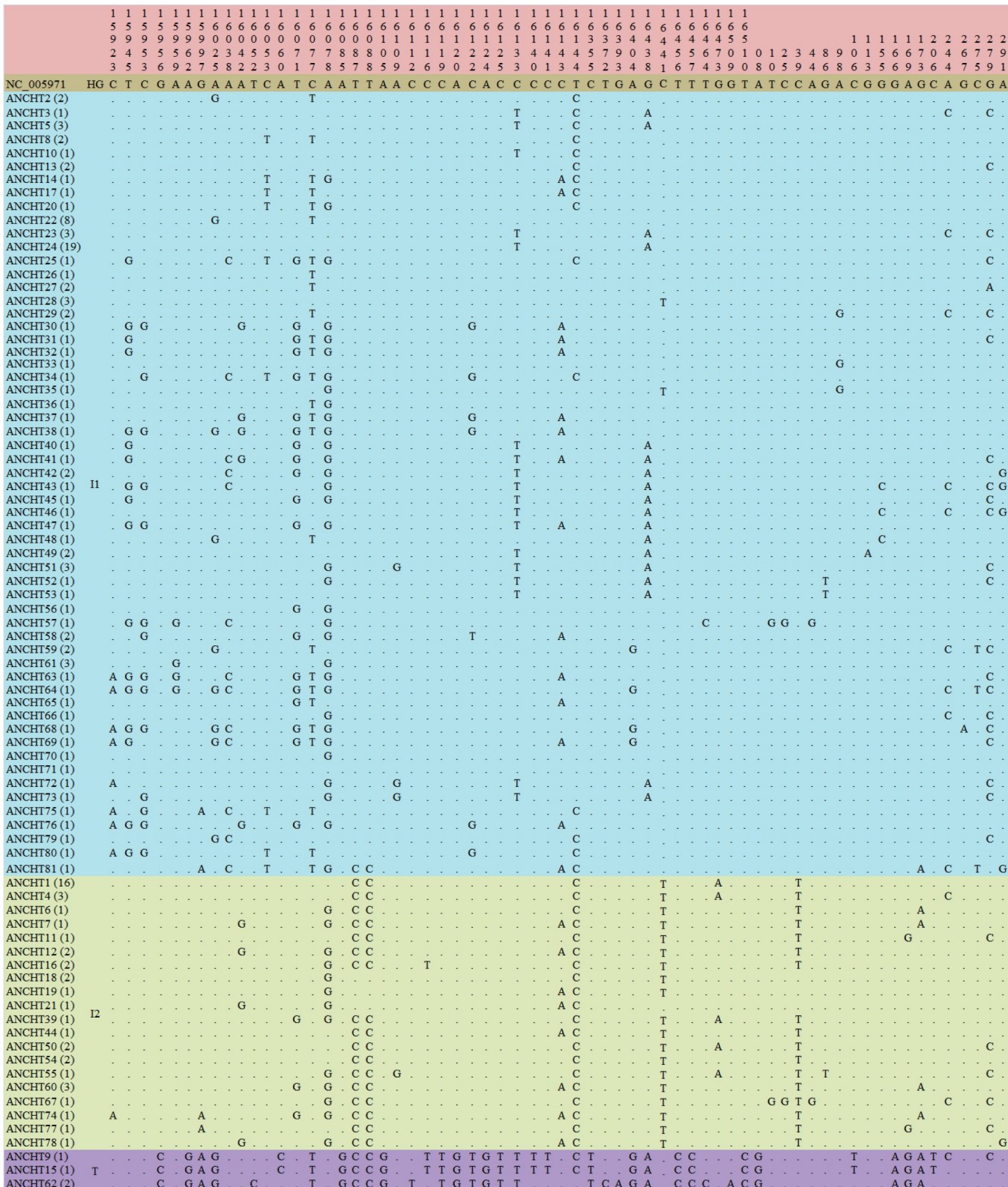

**Fig 2. Sequence variation observed among Andaman cattle haplotypes (ANCHTs) with *Bos indicus* (Zebu cattle) reference sequence (NC_005971) [46].** The sequence information of ANCHTs were aligned with NC_005971 using 913 bp sequence information (bp 15795–16341,1–366). Only variable sites are shown. The sequence positions of every variable sites are indicated above. Dots (.) indicate identity with the reference sequence and different base letters denote substitution. HG indicates Haplogroup. Haplogroups were traced by anunrooted neighbor-joining tree (Fig 3).

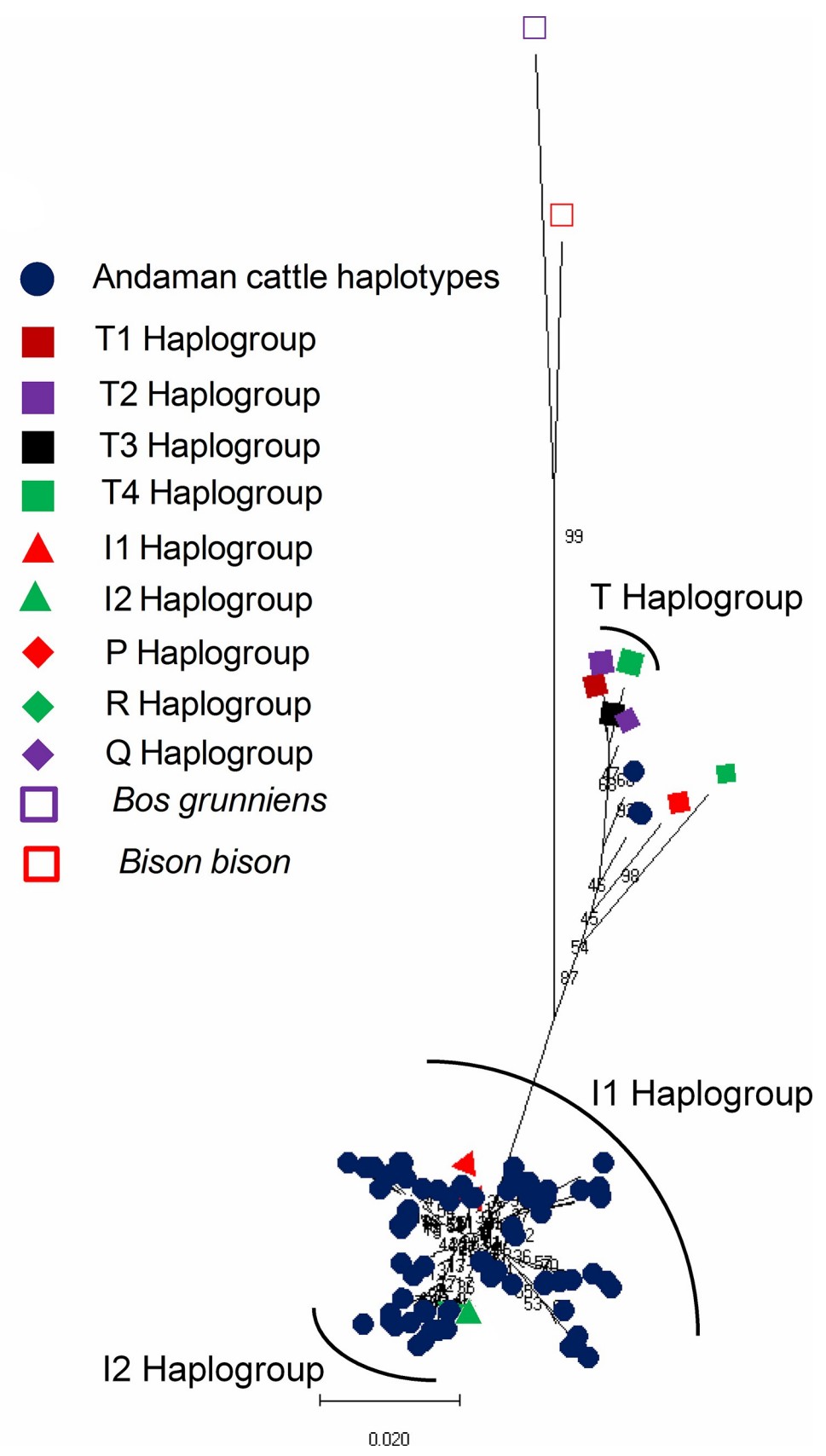

**Fig 3. Haplogroup assignment of different haplotypes of Andaman cattle (ANC).** (a) Neighbor Joining (NJ) phylogenetic tree of ANC haplotypes with standard cattle haplogroups. The GenBank accession numbers of the standard haplogroups used are as follows; I1 = L27722 [49] and AB268579, I2 = EU177869 [50] and AB268559, T1 = LC013968 [51], T2 = AB117049 [52], T3 = V00654 [47], T4 = LC013966 [51], R = HQ184045 [53], P = DQ124389 [50], Q = EU177867 [50]. GQ464312 for *Bos grunniens* [54] and EU177871 for *Bison bison* [50] were used as outgroups. NJ tree was drawn in MEGAX [36].

haplogroup. A haplotype network map of Andaman cattle haplotypes belonging to I haplogroup is presented in S3 Fig.

Phylogeny (Fig 4A) and network (Fig 4B) analysis of the three ANC haplotypes (ANCHT9, ANCHT15 and ANCHT62) which did not fall under I haplogroup with standard haplogroups indicated that they belonged to a cluster comprising of T and Q haplogroups.

## Haplotype I1 and its relationship with I1 haplotypes cattle of Island Southeast Asian countries and Indian subcontinent

Among all haplotypes of ANC, a total of 58 haplotypes belonged to haplogroup I1. The NJ based phylogenetic tree and Baysian relationship of the Andaman cattle haplotypes with cattle sequences (of I1 haplogroup) of different regions representing Indian subcontinent and Island Southeast Asia are presented in Fig 5A and 5B respectively. Poor phylogeographic signal among the animals of different regions was observed. ANCHT2, ANCHT8, ANCHT17, ANCHT20, ANCHT25, ANCHT34, ANCHT75, ANCHT80 and ANCHT81 were phylogenetically close to Southern India and Western India. ANCHT22, ANCHT26-27 and ANCHT29 fell in the cluster comprising of Bhutan, Central India, Western India, Myanmar, Thailand, Southern India and Central India. ANCHT59 was phylogenetically close to Bhutan. ANCHT28, ANCHT30-32, ANCHT35-38, ANCHT56-58, ANCHT61, ANCHT63-66, ANCHT68-69, ANCHT70 and ANCHT76 fell in the same cluster with Western India. ANCHT3, ANCHT5, ANCHT10, ANCHT23-24, ANCHT40-43, ANCHT45-47, ANCHT49, ANCHT51-53, ANCHT72-73 were close to sequences of Western India and Bhutan, ANCHT13 and ANCHT79 were close to Thailand, Southern India, Western India, Bhutan and Central India, ANCHT48 was close to Bhutan and Central India. ANCHT33 and ANCHT71 were in the same cluster with sequences from Bhutan, Myanmar, Vietnam, Philippines and Western India. A network profile of the representative sequences belonging to haplogroup I1 is presented in Fig 5C.

We calculated the pairwise FST distances among I1 sequences of different regions to understand the genetic differentiation (Table 1). The FST values ranged from -0.00737 to 0.18486. When Andaman cattle was compared with cattle of different regions, the FST value was found lowest with Nepal (FST = 0.07712) and highest with Philippines (FST = 0.18741). Moreover, FST values between Andaman cattle with Thailand (0.12242), Central India (0.12287), Southern India (0.13835), Myanmar (0.14425), Eastern India (0.14880) and Bhutan (0.14728) were found lower as compared to those of the other regions. An analysis of molecular variance (AMOVA) indicated 11.72% variation lies among population and 88.28% within population (Table 2).

## Haplogroup I2 and its relationship with I2 cattle of Island Southeast Asian countries and Indian subcontinent

In the present study, a total of 20 haplotypes of ANC fell in haplogroup I2 of *Bos indicus*. NJ phylogenetic tree and Baysian relationship of the ANC haplotypes with cattle sequences of different regions belonging to I2 haplogroups are presented in Fig 6A and 6B respectively. No region-specific clusters were observed. ANCHT18-19 and ANCHT21 formed cluster with Pakistan, ANCHT16 fell in a cluster comprising of Eastern India, Central India, Western and

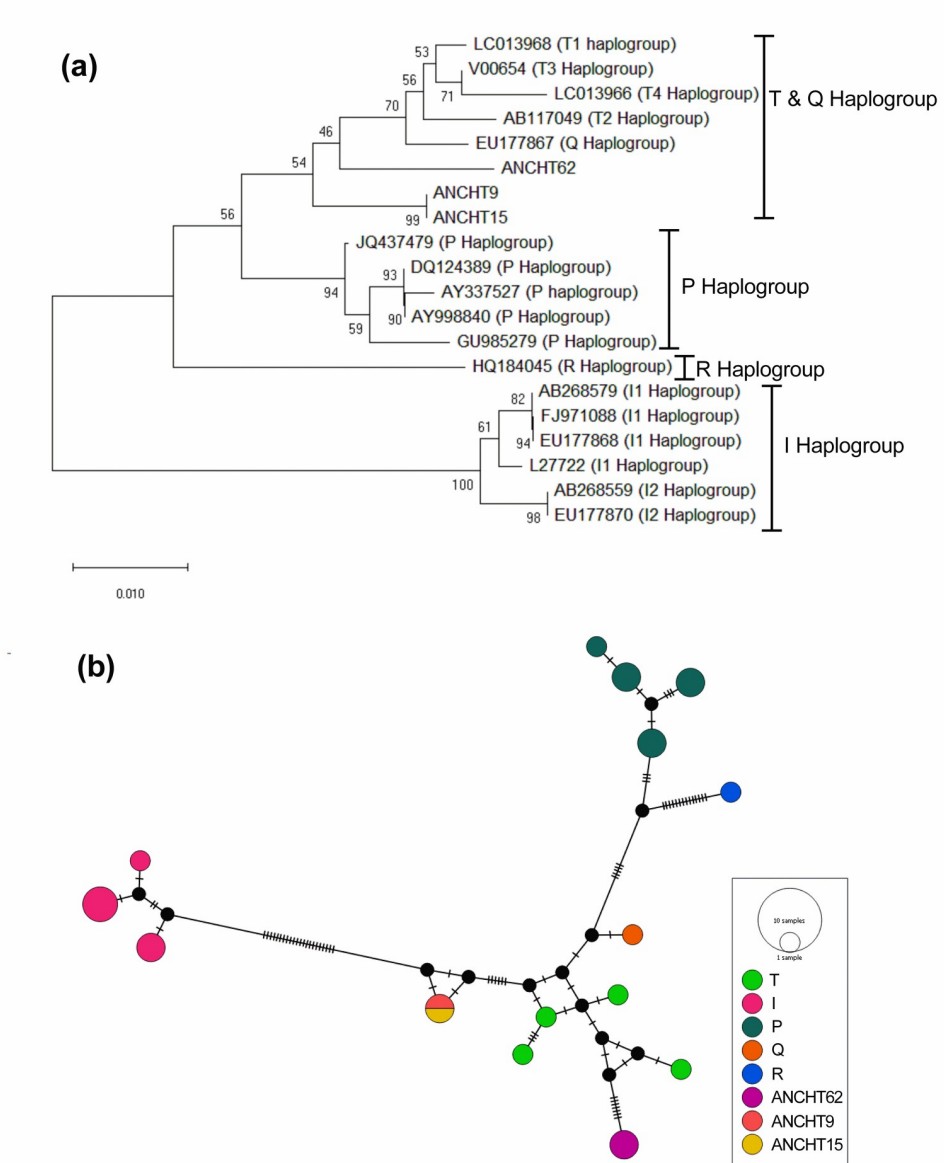

**Fig 4. Phylogeny and network map of three Andaman cattle haplotypes (ANCHT9, ANCHT15 and ANCHT62).**
(a) Neighbor Joining (NJ) phylogenetic tree of the haplotypes with standard cattle haplogroups, (b) A haplotype network map of the haplotypes. The NJ tree was drawn in MEGAX [36] and the network was drawn in PopART ver. 1.7 [42].

Southern India, Thailand and Bhutan. ANCHT1, ANCHT4 and ANCHT50 formed cluster with Southern and Central India. ANCHT6-7, ANCHT11-12, ANCHT39, ANCHT44, ANCHT55, ANCHT60, ANCHT67, ANCHT74 and ANCHT78 were in the same cluster with Eastern India, Bhutan, Nepal, Myanmar and Southern India. ANCHT54 and ANCHT77 were in same cluster with Eastern India and Western India. The median joining network profile of the I2 haplotypes is presented in Fig 6C. Two star shaped phylogenetic network was observed, one comprising of Southern India, Myanmar, Western India, Philippines, Thailand, Vietnam, Eastern India, Nepal and the other of Andaman, Bangladesh, Bhutan, Central India, Eastern India, Southern India, Western India, Myanmar, Nepal, Pakistan, Philippines and Thailand.

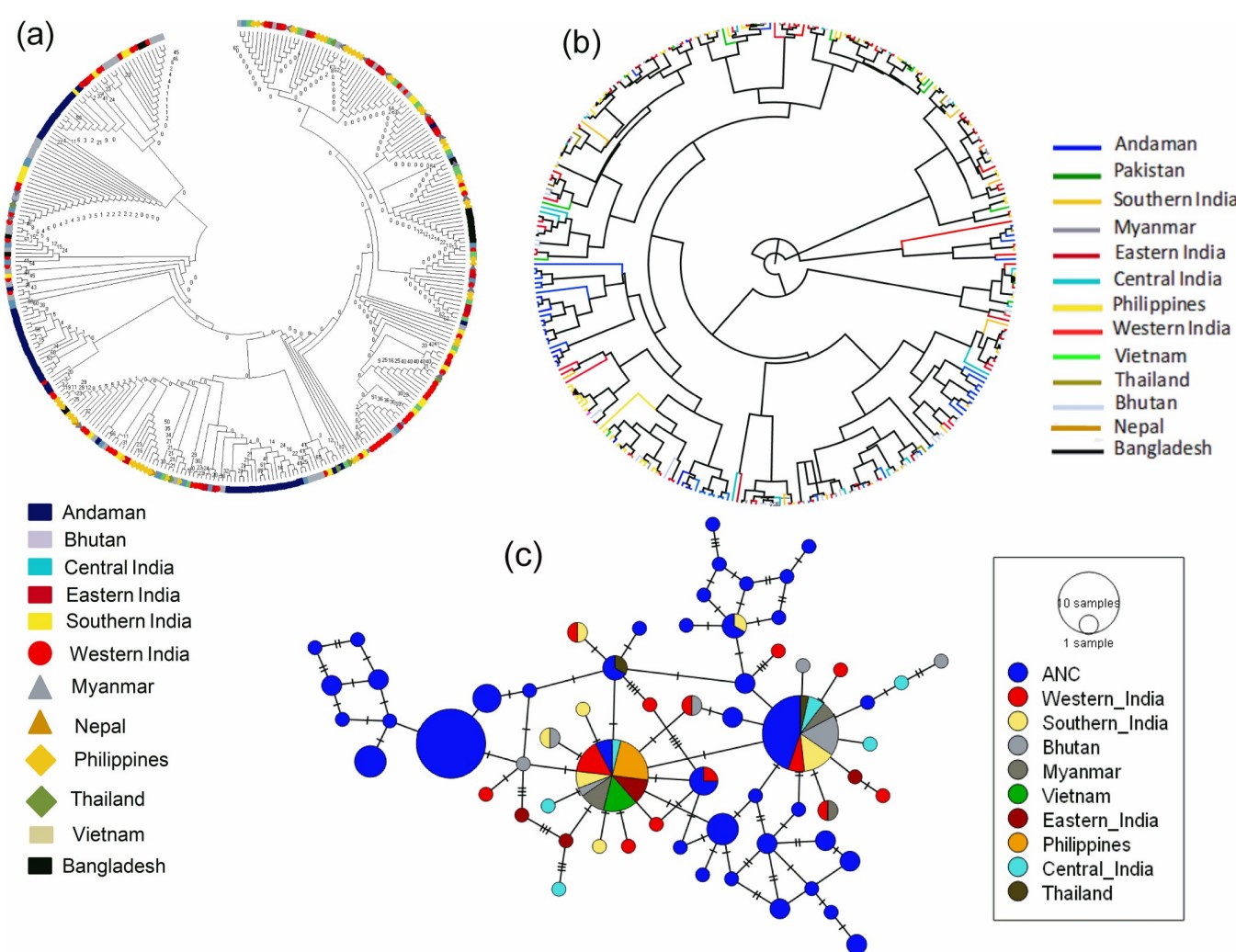

**Fig 5. Haplotype I1 of ANC and its relationship with I1 cattle of Island Southeast Asian countries and Indian subcontinent.** (a) Neighbor Joining (NJ) phylogenetic tree, (b) Bayesian phylogenetic tree, (c) A haplotype network map of the haplotypes.

To calculate the genetic differentiation among the I2 haplogroup cattle of Andaman and different regions of Indian subcontinent and Island Southeast Asia, we calculated the pairwise FST distances (Table 3). Pairwise FST distances among the groups ranged from -0.00635 to 0.72243. When Andaman cattle was compared with cattle of different regions, the FST value was found lowest with Western India (FST = 0.13094) and highest with Vietnam (FST = 0.67755). Moreover, Andaman cattle was found closely related to cattle of Bhutan (0.14356), Central India (0.18030) and Nepal (0.20088) in terms of Pairwise FST values. An analysis of molecular variance (AMOVA) indicated 17.35% variation among population and 82.65% variation within population (Table 4).

## Haplogroup T and its relationship with T haplogroup cattle of Island Southeast Asian countries and Indian subcontinent

NJ based phylogenetic tree of Andaman cattle haplotypes belonging to haplogroup T with T haplogroups of different regions is presented in Fig 7A. Phylogenetic analysis revealed two major clades and Andaman cattle haplotypes share clades with Nepal, Central India, Eastern

**Table 1. Pairwise FST values of I1 cattle of Andaman and different regions of Indian subcontinent and Island Southeast Asian countries.**

| | 1 | 2 | 3 | 4 | 5 | 6 | 7 | 8 | 9 | 10 | 11 | 12 |
|---|---|---|---|---|---|---|---|---|---|---|---|---|
| 1. Andaman | 0.00000 | | | | | | | | | | | |
| 2. Bangladesh | 0.18486* | 0.00000 | | | | | | | | | | |
| 3. Bhutan | 0.14728* | 0.04508* | 0.00000 | | | | | | | | | |
| 4. Central India | 0.12287* | 0.08627* | -0.01295 | 0.00000 | | | | | | | | |
| 5. Eastern India | 0.14880* | 0.10048* | 0.02620 | 0.01401 | 0.00000 | | | | | | | |
| 6. Southern India | 0.13835* | 0.04801 | -0.01428 | -0.00556 | 0.02368 | 0.00000 | | | | | | |
| 7. Western India | 0.15534* | 0.05806* | 0.01119 | 0.00888 | 0.00437 | -0.00606 | 0.00000 | | | | | |
| 8. Myanmar | 0.14425* | 0.07798* | 0.01191 | 0.00821 | -0.00737 | 0.00713 | -0.01112 | 0.00000 | | | | |
| 9. Nepal | 0.07712 | 0.00900 | 0.02455 | 0.03171 | 0.05374 | 0.02392 | 0.00315 | 0.06377 | 0.00000 | | | |
| 10. Philippines | 0.18741* | 0.16021* | 0.10307* | 0.11291* | 0.05130* | 0.11497* | 0.05681* | 0.04367* | 0.02342 | 0.00000 | | |
| 11. Thailand | 0.12242* | 0.14099* | 0.02641 | 0.00768 | 0.03723 | -0.00485 | -0.01221 | 0.01801 | 0.06383 | 0.13634* | 0.00000 | |
| 12. Vietnam | 0.16399* | 0.15903* | 0.06422* | 0.05698* | 0.01544 | 0.07057* | 0.02648* | 0.02004 | 0.10938 | 0.04192 | 0.04657 | 0.00000 |

* indicates p value ≤ 0.05.

India and Bhutan. Bayesian based phylogenetic tree (Fig 7B) indicated that Andaman cattle formed separate clade and was close to sequences of Nepal, Central India, eastern India and Bhutan. The median joining haplotype network of the T haplogroup cattle is presented in Fig 7C; Andaman cattle showed closed proximity to T haplogroup cattle of Nepal, Central India, Eastern India and Bhutan.

Genetic differentiation among the T haplogroup cattle of Andaman and different regions of Indian subcontinent and Island Southeast Asia was estimated based on Pairwise FST distance (Table 5). It was found that FST values ranged from -0.00251 to 0.68783. Andaman cattle had lowest FST value with Eastern India (0.53310) and highest with Western India (0.68783). The FST value of Andaman cattle with those of Central India (0.53846) was very close to the value of ANC vs Eastern India. AMOVA result revealed that 29.92% variation lies among population and 70.08% within population (Table 6).

## Population dynamics parameters of ANC

To infer population demographic history of the two predominant maternal haplogroups (I1 and I2) Andaman cattle, mismatch distribution and neutrality tests were calculated. Mismatch distribution maps of the haplogroups I1 and I2 (Fig 8) were unimodal in nature indicating recent population expansion events [55] for both the haplogroups. Neutrality test results (Tajima's D test, Fu's FS test, Fu and Li's D test, Fu and Li's F test) indicated negative significant Fu's FS value for both I1 and I2 haplogroups of Andaman cattle. Values of Tajima's D test, Fu and Li's D test, Fu and Li's F test were negative for both I1 and I2 but were not significant.

## Discussion

The attenuation and erosion of animal genetic resources especially indigenous domestic breeds is a major global concern as many indigenous breeds became extinct mainly due to

**Table 2. AMOVA analysis of I1 cattle of Andaman and different regions of Indian subcontinent and Island Southeast Asian countries.**

| Source of variation | d. f. | Sum of squares | Variance Components | Percentage of variation | Fixation Index (FST) | P-value |
|---|---|---|---|---|---|---|
| Among populations | 11 | 66.606 | 0.17112 Va | 11.72 | 0.11724 | 0.00000±0.00000 |
| Within populations | 350 | 450.968 | 1.28848 Vb | 88.28 | | |
| Total | 361 | 517.575 | 1.45961 | | | |

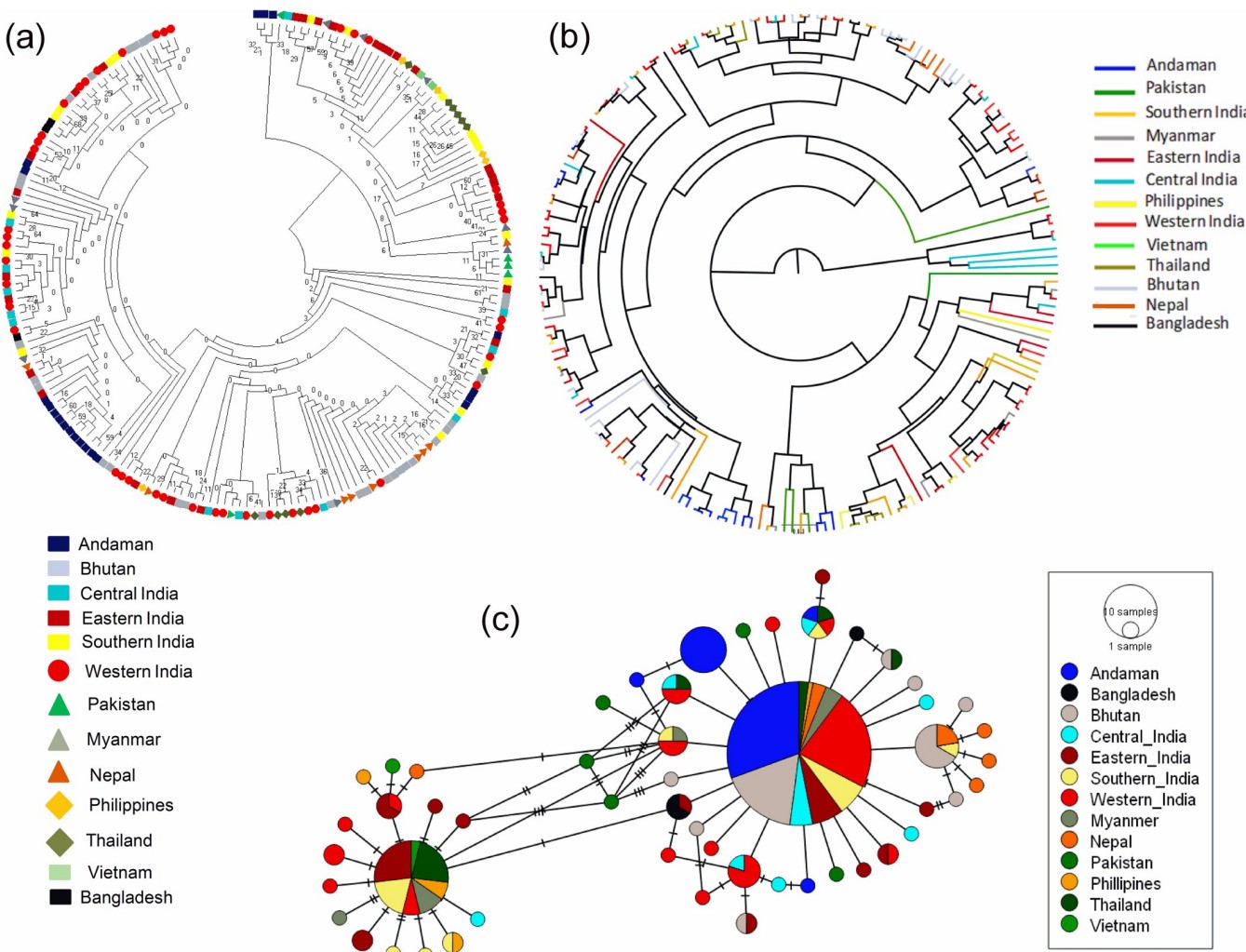

**Fig 6. Haplogroup I2 and its relationship with I2 haplogroup cattle of Island Southeast Asian countries and Indian subcontinent.** (a) Neighbor Joining (NJ) phylogenetic tree, (b) Bayesian phylogenetic tree, (c) A haplotype network map of the haplotypes.

crossbreeding with exotic breeds [56]. Conservation of animal genetic resources (AnGR) is extremely important for maintenance of global biodiversity and has been stressed upon as one of the major strategic priority areas of global plan of action of AnGR [57]. Effective management of AnGR necessitates knowledge on molecular genetic structure of breeds [58, 59]. Moreover, for conservation and development of breeding strategy, accurate information on breed genetic structure including genetic diversity, phylogenetic relationship with other cattle germplasm and evolutionary history of the breed is very much necessary. Andaman cattle distributed in small pockets of Andaman and Nicobar Islands are zebu type and very much adapted to the hot humid climate of Andaman and Nicobar Islands and thrive well with low management. Due to introduction of exotic breeds for genetic up-gradation of the indigenous breed, the breed is highly threatened and needs immediate attention and conservation efforts. Information on genetic structure including genetic diversity and phylogeography of the breed is necessary for formulation of a suitable breeding plan and conservation strategy of the breed [15]. Mitochondrial control region has been used extensively as an optimal marker to understand genetic diversity, phylogeny, phylogeography of a breed and evolutionary relationship

**Table 3. Pairwise FST values of I2 cattle of Andaman and different regions of Indian subcontinent and Island Southeast Asian countries.**

|  | 1 | 2 | 3 | 4 | 5 | 6 | 7 | 8 | 9 | 10 | 11 | 12 |
|---|---|---|---|---|---|---|---|---|---|---|---|---|
| 1. Andaman | 0.00000 |  |  |  |  |  |  |  |  |  |  |  |
| 2. Bhutan | 0.14356* | 00000 |  |  |  |  |  |  |  |  |  |  |
| 3. Central India | 0.18030* | 0.13031* | 0.00000 |  |  |  |  |  |  |  |  |  |
| 4. Eastern India | 0.30860* | 0.29139* | 0.13968* | 0.00000 |  |  |  |  |  |  |  |  |
| 5. Southern India | 0.28083* | 0.26621* | 0.12002* | -0.00635 | 0.00000 |  |  |  |  |  |  |  |
| 6. Western India | 0.13094* | 0.08739* | 0.05553* | 0.09285* | 0.06218* | 0.00000 |  |  |  |  |  |  |
| 7. Myanmar | 0.32781* | 0.33058* | 0.10856* | -0.04568 | -0.04816 | 0.05602 | 0.00000 |  |  |  |  |  |
| 8. Nepal | 0.20088* | 0.04284 | 0.08776* | 0.20153* | 0.16815* | 0.09213* | 0.20929* | 0.00000 |  |  |  |  |
| 9. Pakistan | 0.46680* | 0.48536* | 0.18317* | 0.09338 | 0.10946 | 0.22522* | 0.07297 | 0.29677* | 0.00000 |  |  |  |
| 10. Philippines | 0.53906* | 0.55671* | 0.23326* | 0.03331 | 0.05575 | 0.27310* | 0.02484 | 0.35322* | 0.07143 | 0.00000 |  |  |
| 11. Thailand | 0.34813* | 0.36125* | 0.15119* | -0.02441 | -0.04968 | 0.07702* | -0.06326 | 0.25338* | 0.10433 | 0.03354 | 0.00000 |  |
| 12. Vietnam | 0.67755* | 0.72243* | 0.32233* | 0.05514 | 0.14932 | 0.39368* | 0.13848 | 0.55870* | 0.11521 | -0.21872 | 0.18656 | 0.00000 |

* indicates p value ≤ 0.05

among breeds [15, 60, 61]. Here, we have studied the population genetic structure and genetic diversity of Andaman cattle and their evolutionary relationship with indigenous cattle distributed across Indian subcontinent and Island Southeast Asia.

The nucleotide composition of complete mitochondrial D-loop indicated bias towards A+T nucleotides as compared to G+C which is a common feature of mitochondrial genomes of vertebrates [62]. Higher percentage of A+T as compared to G+C was also reported in complete D-loop sequences of Aceh cattle of Indonesia [63] and Chinese cattle breeds [34, 64]. Positive AT skewness and negative GC skewness indicated that A and C occurred more frequently than T and G respectively. In the present study, the rate of transition was found higher than that of transversion which is considered as a unique feature of mtDNA D-loop sequences of human and animals [65–67]. Polymorphism analysis indicated 64 polymorphic or variable sites in Andaman cattle with nucleotide diversity ($\pi \pm SD$) of 0.00952 ± 0.00076. Haplotype analysis detected 81 haplotypes with haplotype diversity of 0.968 ± 0.008. The haplotype diversity of Andaman cattle was found higher than those of Eurasian cattle breeds (0.746 ± 0.029) [68] and Chinese cattle breeds (0.904 ± 0.008) [69]. High genetic diversity in Andaman cattle indicated high polymorphisms in mtDNA D-loop region and very low level of inbreeding. High genetic diversity of Andaman cattle is also indicative of gene flow from other cattle breeds [70]. Andaman cattle are generally reared in free range system of rearing and mating happens naturally in which there is a high chance of gene flow from other breeds. Moreover, no directional or selective breeding is being practised in Andaman cattle; that might be the reason behind high genetic diversity of Andaman cattle as directional breeding for a production trait reduces genetic diversity [71–73]. In addition, rich genetic diversity of Andaman cattle indicates its high ability to adapt to environmental changes as breeds with high genetic diversity have stronger ability to adjust to any changes in the environment as compared to breeds

**Table 4. AMOVA analysis of I2 cattle of Andaman and different regions of Indian subcontinent and Island Southeast Asian countries.**

| Source of variation | d.f. | Sum of squares | Variance Components | Percentage of variation | Fixation Index (FST) | P-value |
|---|---|---|---|---|---|---|
| Among populations | 11 | 83.028 | 0.35783 Va | 17.35 | 0.17350 | 0.00000±0.00000 |
| Within populations | 195 | 332.388 | 1.70455 Vb | 82.65 |  |  |
| Total | 206 | 415.415 | 2.06238 |  |  |  |

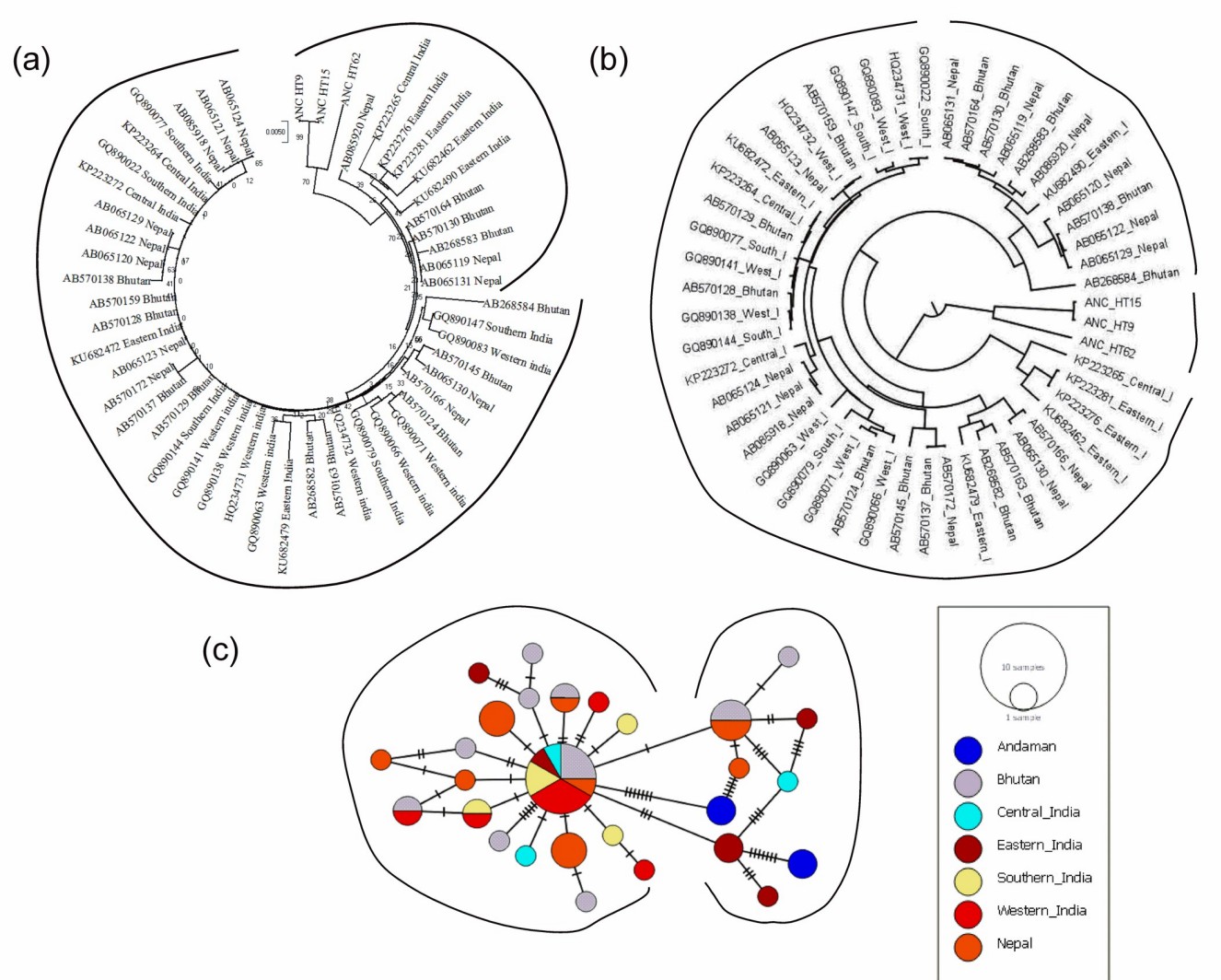

**Fig 7. Haplogroup T and its relationship with T haplogroup cattle of Island Southeast Asian countries and Indian subcontinent.** (a) Neighbor Joining (NJ) phylogenetic tree, (b) Bayesian phylogenetic tree, (c) A haplotype network map of the haplotypes.

**Table 5. Pairwise FST values of T cattle of Andaman and different regions of Indian subcontinent and Island Southeast Asian countries.**

|  | 1 | 2 | 3 | 4 | 5 | 6 | 7 |
|---|---|---|---|---|---|---|---|
| 1. Andaman | 0.00000 |  |  |  |  |  |  |
| 2. Bhutan | 0.64539* | 00000 |  |  |  |  |  |
| 3. Central India | 0.53846* | -0.00251 | 0.00000 |  |  |  |  |
| 4. Eastern India | 0.53310* | 0.14935* | -0.06375 | 0.00000 |  |  |  |
| 5. Southern India | 0.68664* | -0.04048 | 0.09263 | 0.17148 | 0.00000 |  |  |
| 6. Western India | 0.68783* | -0.00512 | 0.10339 | 0.19613* | -0.09483 | 0.00000 |  |
| 7. Nepal | 0.65853* | 0.01019 | 0.05482 | 0.20001* | 0.03439 | 0.06496 | 0.00000 |

* indicates p value $\leq$ 0.05.

**Table 6. AMOVA analysis of T cattle of Andaman and different regions of Indian subcontinent and Island Southeast Asian countries.**

| Source of variation | d.f. | Sum of squares | Variance Components | Percentage of variation | Fixation Index (FST) | P-value |
|---|---|---|---|---|---|---|
| Among populations | 6 | 38.747 | 0.68378 Va | 29.92 | 0.29923 | 0.00000±0.00000 |
| Within populations | 45 | 72.060 | 1.60134 Vb | 70.08 | | |
| Total | 51 | 110.808 | 2.28512 | | | |

with low genetic diversity [74, 75]. Maintaining the genetic diversity of the indigenous cattle breed adapted to the local microenvironment is very important and should be included in conservation strategy.

In domestic cattle, two major mtDNA lineages have been described; taurine (*Bos taurus*) and zebu (*Bos indicus*). *Bos taurus* was further sub-divided into different sub-haplogroups (T1-T5) and *Bos indicus* into two sub-haplogroups (I1 and I2) [13, 14, 50, 76]. Haplogroup T1 is found in African cattle [18], T2 is found in low frequency in cattle of South Europe and Asia [50, 77] and T3 is concentrated in Central and Northern European cattle [18]. North and East Asia cattle harbour T4 haplogroup [52] and T5 haplogroup is detected only in Italy [53] and Croatia [77]. Haplogroup I has been reported to be entered in Indus valley and neighbouring regions [14]. Besides two major haplogroups, five minor haplogroups (P, Q, R, C and E) have been demonstrated. Haplogroup Q is closely related to haplogroup T and is reported in ancient domestic cattle [53, 78] and modern cattle of Eurasia and Africa [50, 76]. Haplogroup R is detected in modern Italian cattle [53]. Haplogroups C and E have been detected in ancient aurochs of China and Germany respectively [79, 80]. In the present study, all the haplotypes belonged to I haplogroup except three haplotypes which belonged to T/Q haplogroup. Close phylogenetic relationship between haplogroup T and haplogroup Q has been reported and it was found that D-loop motif of haplogroup Q is poorly differentiated from haplogroup T [76]. On the other hand, presence of Q haplogroup in cattle population is extremely rare and it is mainly found in ancient cattle population [53, 76, 78]. Though, 2.66% Andaman cattle falls under T/Q haplogroup, it is highly probable that the cattle population harbours T haplogroup. The results of the study indicated that Andaman cattle originated from zebu cattle and presence of T haplogroup though at low frequency indicated taurine gene introgression. In Andaman and Nicobar Islands, female animals of taurine breeds (Jersey, Holstein Friesian) were imported for genetic improvement of Andaman cattle by cross breeding programme. That might be the reason behind presence of T haplogroup in Andaman cattle. Although due care was taken to include Andaman cattle based on phenotypic characteristics and the animals selected appeared morphologically as *Bos indicus*, but as genetics are not always reflected by particular phenotype [81], phenotypically zebuine animals with taurine mitochondria were detected.

Andaman cattle was dominated by I1 haplogroup followed by I2 haplogroup. Andaman cattle of both I1 and I2 haplogroup showed poor phylogeographic signal with genetic affinities to cattle of Indian subcontinent and Island Southeast Asia. However, as per Indian cattle breeds are concerned, Andaman cattle showed close phylogenetic relationship with Red Sindhi, Sahiwal and Ongole breeds of mainland India. No region-specific clustering of cattle was observed. Moreover, AMOVA analysis of I1 and I2 haplogroups indicated that majority of the genetic variation existed within population rather than among population which is indicative of weak phylogenetic structure of Andaman cattle. The poor phylogenetic structure might be due to multidirectional gene flow from different regions or admixture of Andaman cattle with cattle population of different regions. The admixture of Andaman cattle with cattle population of Indian subcontinent and Island South east Asian countries can be explained, because there used to be gene flow or immigration between them. Modern zebu cattle are thought to

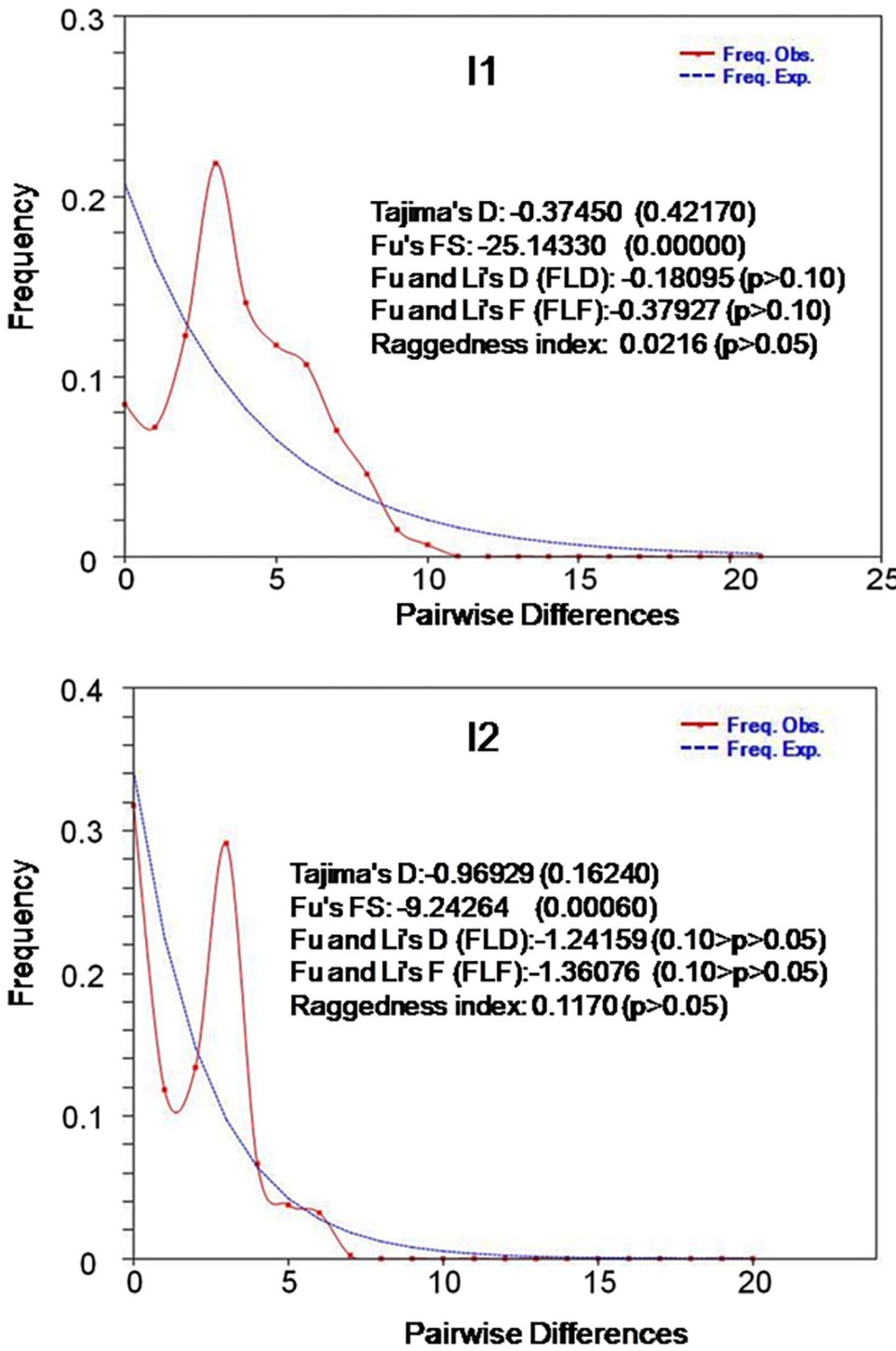

**Fig 8. Mismatch distribution graph and neutrality tests of Andaman cattle.** The x and y axis present number of pairwise differences and the relative frequencies of pairwise comparisons respectively. Mismatch distribution graph was drawn in DnaSP v 6 [38] and neutrality tests were calculated in DnaSP v 6 [38] and Arlequin v 3.5 [45].

have domesticated from South Indian aurochs, *Bos primigenius nomadicus* in Indus valley in Neolithic times [82–84]. Archaeological evidences from Harappa and Mohenjo-Daro civilization established the existence of zebu cattle throughout the Indus valley region ~5000 YBP [85, 86]. Later, South India, Gujarat and Ganges region were proposed as potential centres of zebu domestication [85, 87, 88]. Based on haplotype coalesce analysis, zebu cattle have been classified into two haplogroups; I1 and I2 [67, 89]. Chen *et al.* [14] readdressed the issue and indicated greater Indus valley as the domestication centre of I1 haplogroup whereas the exact domestication centre of I2 haplogroup could not be pinpointed. From the centre of domestication, zebu cattle spread to different areas of Indian subcontinent and Southeast Asia [14, 28] through pastoralism and commercial trade. Cattle probably of I1 haplogroup spread to South China and Southeast India around 1500–1000 BC through pastoralism [90] whereas I2 haplogroup were incorporated into the domestic pool later time due to commercial trade [14]. Varied selection pressure together with genetic drift led to development of different breeds with several phenotypic traits [91, 92].

Archaeological data indicates repeated waves of migration from Indian subcontinent to Southeast Asian countries in Neolithic age and cattle accompanied human migrations [22]. In addition, historical evidence on maritime trade indicated trade between Austronesian people of Island South east Asia with Southern India as early as 1500 BC [93] through sea route and Indian Ocean was the natural corridor [94]. Kingdoms of Indian subcontinent established political and cultural influences on Southeast Asian kingdoms in Burma, Thailand, Indonesia, Philippines, Cambodia and Champa as early as 290 BC [23, 24]. During the ancient time, maritime trade was associated with halt at intermediary places for replenishment. Andaman and Nicobar Islands, an ideal natural harbour, was used as halting points [95, 96]. It is tantalizing to hypothesize that this multidirectional diffusion shapes the genetic structure of Andaman cattle. High genetic diversity of Andaman cattle supports this hypothesis as high genetic diversity is associated with mixing of populations from different geographical regions [97]. Moreover, study on peopling of Andaman and Nicobar Islands indicated the origin of Andaman islanders from Southeast Asian countries (Malaysia, Thailand) and China [98–100]. Most probably, cattle accompanied human migrations and acted as the founding population of Andaman cattle. With the advent of colonization, the colonial people including the Danish, Austrian, Japanese and British who ruled Andaman and Nicobar Islands from 1755 to 1947 introduced several livestock and poultry breeds from Southeast Asian countries [101, 102]. These breeds might have contributed in shaping the genetic structure of Andaman cattle. A similar case is found in Trinket cattle of Nicobar group of islands, which has been domesticated from Island Southeast Asia [103, 104]. Therefore, origin of Andaman cattle recapitulates the legacies of Neolithic diffusion from centre of domestication of zebu cattle along with multidirectional commercial exchange between Indian subcontinent and Island Southeast Asia.

The unimodal mismatch distribution pattern of I1 and I2 haplogroups of Andaman cattle indicates recent population expansion events [55]. However, the NJ network of the haplotypes of ANC (Figs 5–7) does not show a star like pattern which is considered as a hallmark of population expansion. Moreover, non-significant neutrality test values including Tajima's D test, Fu and Li's D test, Fu and Li's F test do not support population expansion. Most probably, short term demographic expansion or absence of enough mutation in the gene led to the confusing results [72]. The genetic differentiation of Andaman cattle in the present study might be due to reproductive isolation of the breed [105] owing to geographical separation of ANI from rest of the world.

This is the first insight into the genetic history of Andaman cattle and its evolutionary relationship with indigenous cattle of Indian subcontinent and Island Southeast Asia. The results showed that Andaman cattle had maternal origin of *Bos indicus* and experienced gene flow

from different regions and has intriguing implications for the history of human movements and commerce. Therefore, origin of Andaman cattle recapitulates the legacies of Neolithic diffusion from centre of domestication of zebu cattle along with multidirectional commercial exchange between Indian subcontinent and Island Southeast Asia.

## Supporting information

**S1 Table. Details of sampling location of Andaman cattle (ANC).**
(DOCX)

**S2 Table. GenBank Accession numbers used in the study.**
(DOCX)

**S3 Table. Haplogroup assignment of Andaman cattle haplotypes.**
(DOCX)

**S1 Fig. Frequency of the detected haplotypes of Andaman cattle.** ANC indicates Andaman cattle and HT indicates haplotypes.
(TIF)

**S2 Fig. Sequence variation observed among Andaman cattle haplotypes (ANCHTs) with *Bos taurus* (Taurine cattle) reference sequence.** The sequence information of ANCHTs were aligned with V00654 using 913 bp sequence information (bp 15792..16338,1..363). Only variable sites are shown. The sequence positions of every variable sites are indicated above. Sequence identities are indicated by dots (.) and different base letters denote substitution. HG indicates Haplogroup. Haplogroups were traced by an unrooted neighbor-joining tree (Fig 3).
(TIF)

**S3 Fig. A haplotype network map of Andaman cattle haplotypes belonging to I haplogroup.** Circle area (node) is proportional to frequency. Dashed lines represent mutations.
(TIF)

## Author Contributions

**Conceptualization:** Arun Kumar De, Debasis Bhattacharya.

**Formal analysis:** Arun Kumar De, Sneha Sawhney, Ramachandran Muthiyan, Asit Kumar Bera.

**Funding acquisition:** Arun Kumar De.

**Investigation:** Arun Kumar De, Sneha Sawhney, Ramachandran Muthiyan, Dhruba Malakar, Samiran Mondal.

**Methodology:** Arun Kumar De, Sneha Sawhney, Perumal Ponraj, Jai Sunder, T. Sujatha, Ashish Kumar, P. A. Bala.

**Project administration:** Arun Kumar De.

**Software:** Arun Kumar De.

**Validation:** Arun Kumar De.

**Writing – original draft:** Arun Kumar De.

**Writing – review & editing:** Debasis Bhattacharya, Eaknath B. Chakurkar.

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
