## [Decision Letter · Decision Letter 0]

27 Sep 2022

PONE-D-22-11680Legacies of domestication, Neolithic diffusion and trade between Indian subcontinent and Island Southeast Asia shape maternal genetic diversity of Andaman cattlePLOS ONE

Dear Dr. De,

Thank you for submitting your manuscript to PLOS ONE. After careful consideration, we feel that it has merit but does not fully meet PLOS ONE’s publication criteria as it currently stands. Therefore, we invite you to submit a revised version of the manuscript that addresses the points raised during the review process.

As you can see that the eminent reviewer is happy with the study design. However, some changes are needed before we make final decision. Therefore, please consider the comments and revise the manuscript accordingly.

We look forward to receiving your revised manuscript.

Kind regards,

Gyaneshwer Chaubey

Academic Editor

PLOS ONE

Journal Requirements:

2. In your Methods section, please provide additional details regarding participant consent from the owners of the animals. In the ethics statement in the Methods and online submission information, please ensure that you have specified (1) whether consent was informed and (2) what type you obtained (for instance, written or verbal). If the need for consent was waived by the ethics committee, please include this information.

 “This work was supported by a Grant from Indian Council of Agricultural Research, New Delhi, India (Grant No. HORTCIARISIL201700800181). grant recipient AKD”

4. We note that Figure 9in your submission contain [map/satellite] images which may be copyrighted. All PLOS content is published under the Creative Commons Attribution License (CC BY 4.0), which means that the manuscript, images, and Supporting Information files will be freely available online, and any third party is permitted to access, download, copy, distribute, and use these materials in any way, even commercially, with proper attribution. For these reasons, we cannot publish previously copyrighted maps or satellite images created using proprietary data, such as Google software (Google Maps, Street View, and Earth). For more information, see our copyright guidelines: http://journals.plos.org/plosone/s/licenses-and-copyright.

 a. You may seek permission from the original copyright holder of Figure 9 to publish the content specifically under the CC BY 4.0 license. 

Reviewers' comments:

Reviewer's Responses to Questions

**Comments to the Author**

1. Is the manuscript technically sound, and do the data support the conclusions?

Reviewer #1: Yes

2. Has the statistical analysis been performed appropriately and rigorously? 

Reviewer #1: Yes

3. Have the authors made all data underlying the findings in their manuscript fully available?

Reviewer #1: Yes

4. Is the manuscript presented in an intelligible fashion and written in standard English?

Reviewer #1: Yes

5. Review Comments to the Author

Reviewer #1: The study carried out has given a new insight into Indian Andaman island cattle, delineating possible migration route using mitochondrial D-loop sequence information. The findings can help to justify the haplogroup distribution in Deccan and coastal cattle breeds' migration routes and population structure.

The manuscript is overall well written, but minor concerns listed below need to be addressed by the authors.

• Describe the phenotypic characteristics of the Andaman cattle and its close resemblance to main land Indian breed if any? Also the homogeneity in the population, given the fact that there has been introgression from other multiple regions.

• Line 454- Haplotype sharing between indicus and taurine cattle is also reported. Authors should specify that reason also.

• In figure S1, 81 haplotypes of Andaman cattle have been presented against the standard cattle haplotypes. The most prominent haplotype according to findings is I2 followed by I2 and T. But in figure 3, the authors didn’t include the I2 reference haplotype in the NJ tree (green triangle indicated in the legend). Alternately, the authors may provide this information (ANCH haplotypes belonging to various standard cattle haplotypes) in supplementary files.

• The Q and T haplogroups have been reported at 2.66 % in the Andaman cattle population. The haplogroup affiliation for the Bos taurus and Bos indicus mtDNA sequences mentioned in table S2 doesn’t indicate any Q haplogroup reference. Justify the Q haplogroup in the studied population.

• The extra legends in Figure 5 and Figure 6 need to be revised.

• Figure 5 legend- check the spelling of ‘subcontinent’

• Data presented in Figure 7 is already part of Figure 6. Delete Figure 7 or justify it.

• Closest breed to ANCH of Indian mainland, based on the results of the study should be mentioned.

• Clarify, whether still introgression of other breeds including taurine from mainland India is still going on in the studied ANC population?

• Delete Table 1. Data is already mentioned in the text.

Minor-changes

Line 529- it should be ‘indicates’

Statement conclusion- ‘The lack of a strong phylogenetic structure in Andaman cattle indicated multidirectional gene flow from different regions’ contradicts the statement in previous paragraph- line 535- regarding strong genetic differentiation of ANC?

6. PLOS authors have the option to publish the peer review history of their article (what does this mean?). If published, this will include your full peer review and any attached files.

Reviewer #1: **Yes: **Dr Ranjit Singh Kataria, Principal Scientist, ICAR-National Bureau of Animal Genetic Resources, Karnal, India

---

## [Author Response · Author response to Decision Letter 0]

20 Nov 2022

Response to Reviewers

PONE-D-22-11680

Legacies of domestication, Neolithic diffusion and trade between Indian subcontinent and Island Southeast Asia shape maternal genetic diversity of Andaman cattle

PLOS ONE

Reviewer's Responses to Questions

1. Is the manuscript technically sound, and do the data support the conclusions?

Reviewer #1: Yes

2. Has the statistical analysis been performed appropriately and rigorously?

Reviewer #1: Yes

3. Have the authors made all data underlying the findings in their manuscript fully available?

Reviewer #1: Yes

4. Is the manuscript presented in an intelligible fashion and written in standard English?

Reviewer #1: Yes

5. Review Comments to the Author

Reviewer #1: The study carried out has given a new insight into Indian Andaman island cattle, delineating possible migration route using mitochondrial D-loop sequence information. The findings can help to justify the haplogroup distribution in Deccan and coastal cattle breeds' migration routes and population structure.

The manuscript is overall well written, but minor concerns listed below need to be addressed by the authors.

Response: The authors would like to thank the reviewer for his enthusiasm in the current manuscript. The minor concerns raised by the reviewer have been addressed. The authors are thankful to the reviewer for his constructive comments for the improvement of the manuscript. 

• Describe the phenotypic characteristics of the Andaman cattle and its close resemblance to main land Indian breed if any? Also the homogeneity in the population, given the fact that there has been introgression from other multiple regions.

Response: Information on phenotypic characteristics of the breeds and its phenotypically related cattle breeds from mainland India have been included in the introduction section of the manuscript. Introgression from multiple regions has happened long back and over the time the breed bas been stabilized in the microenvironment of Andaman and Nicobar Islands. As per current situation, no introgression is happening as import of cattle from mainland India or any other region is completely banned as per state government policy. Currently the cattle breed depicts homogeneity in phenotypic characteristics and we are in a process to register the breed as indigenous breed at NBAGR. 

• Line 454- Haplotype sharing between indicus and taurine cattle is also reported. Authors should specify that reason also.

Response: In Andaman and Nicobar Islands, female animals of taurine breeds were imported for genetic improvement of Andaman cattle by cross breeding programme long back. That might be the reason behind presence of taurine haplogroup in Andaman cattle. We have included that in the discussion portion of the revised manuscript. 

• In figure S1, 81 haplotypes of Andaman cattle have been presented against the standard cattle haplotypes. The most prominent haplotype according to findings is I1 followed by I2 and T. But in figure 3, the authors didn’t include the I2 reference haplotype in the NJ tree (green triangle indicated in the legend). Alternately, the authors may provide this information (ANCH haplotypes belonging to various standard cattle haplotypes) in supplementary files.

Response: I2 reference haplogroup has been included in the revised figure. Moreover, as suggested, the information has been provided as a supplementary table (S3 Table). 

• The Q and T haplogroups have been reported at 2.66 % in the Andaman cattle population. The haplogroup affiliation for the Bos taurus and Bos indicus mtDNA sequences mentioned in table S2 doesn’t indicate any Q haplogroup reference. Justify the Q haplogroup in the studied population.

Response: All taurine cattle are grouped under macro-haplogroup T. Close phylogenetic relationship between haplogroup T and haplogroup Q has been reported and it was found that D-loop motif of haplogroup Q is poorly differentiated from haplogroup T (Achilli et al., 2009). On the other hand, presence of Q haplogroup in cattle population is extremely rare and it is mainly found in ancient cattle population (Bollongino et al., 2006, Achilli et al., 2009, Bonfiglio et al., 2010). Though, 2.66 % Andaman cattle falls under T/Q haplogroup, it is highly probable that the cattle population harbours T haplogroup. Therefore, during analysis of the relationship of Andaman cattle with indigenous cattle population of Indian subcontinent and Island Southeast Asian countries, T haplogroup was considered. This has been included in the revised manuscript.

Achilli A, Bonfiglio S, Olivieri A, Malusà A, Pala M, Hooshiar Kashani B et al. The multifaceted origin of taurine cattle reflected by the mitochondrial genome. PLoS One. 2009;4: e5753. doi: 10.1371/journal.pone.0005753

Bollongino R, Edwards CJ, Alt KW, Burger J, Bradley DG. Early history of European domestic cattle as revealed by ancient DNA. Biol Lett. 2006;2: 155-159. doi: 10.1098/rsbl.2005.0404.

Bonfiglio S, Achilli A, Olivieri A, Negrini R, Colli L, Liotta L, et al. The enigmatic origin of bovine mtDNA haplogroup R: sporadic interbreeding or an independent event of Bos primigenius domestication in Italy? PLoS One. 2010;5: e15760. doi: 10.1371/journal.pone.0015760.

• The extra legends in Figure 5 and Figure 6 need to be revised.

Response: Legends in Figure 5 and 6 have been revised. 

• Figure 5 legend- check the spelling of ‘subcontinent’

Response: Corrected.

• Data presented in Figure 7 is already part of Figure 6. Delete Figure 7 or justify it.

Response: In Figure 6, haplogroup I2 of Andaman cattle and its relationship with I2 haplogroup cattle of Island Southeast Asian countries and Indian subcontinent was depicted. In Figure 7, haplogroup T of Andaman cattle and its relationship with T haplogroup cattle of Island Southeast Asian countries and Indian subcontinent was depicted. These two figures represent two different results, therefore both the figures may be kept. 

• Closest breed to ANCH of Indian mainland, based on the results of the study should be mentioned.

Response: Based on the results of the study, Andaman cattle showed close phylogenetic relationship with Red Sindhi, Sahiwal and Ongole breed of mainland India. This information has been included in the revised manuscript. 

• Clarify, whether still introgression of other breeds including taurine from mainland India is still going on in the studied ANC population?

Response: Currently no introgression of any cattle germplasm either from mainland India or any other region is happening. Andaman and Nicobar Administration has taken strict steps to conserve the indigenous livestock germplasm including cattle. Therefore, strict ban has been imposed for import of any livestock from mainland India or any other region. As Andaman and Nicobar is geographically isolated, illegal import of livestock is also not possible. Therefore, further introgression is not possible under the current scenario.

• Delete Table 1. Data is already mentioned in the text.

Response: Table 1 has been deleted as suggested.

Minor-changes

Line 529- it should be ‘indicates’

Response: done

Statement conclusion- ‘The lack of a strong phylogenetic structure in Andaman cattle indicated multidirectional gene flow from different regions’ contradicts the statement in previous paragraph- line 535- regarding strong genetic differentiation of ANC?

Response: The statement has been revised. Moreover, the confusing results regarding genetic differentiation has been justified in the discussion part.

---

## [Editor Report · Decision Letter 1]

22 Nov 2022

Legacies of domestication, Neolithic diffusion and trade between Indian subcontinent and Island Southeast Asia shape maternal genetic diversity of Andaman cattle

PONE-D-22-11680R1

Dear Dr. De,

We’re pleased to inform you that your manuscript has been judged scientifically suitable for publication and will be formally accepted for publication once it meets all outstanding technical requirements.

Kind regards,

Gyaneshwer Chaubey

Academic Editor

PLOS ONE
---

## [Editor Report · Acceptance letter]

1 Dec 2022

PONE-D-22-11680R1 

Legacies of domestication, Neolithic diffusion and trade between Indian subcontinent and Island Southeast Asia shape maternal genetic diversity of Andaman cattle 

Dear Dr. De:

I'm pleased to inform you that your manuscript has been deemed suitable for publication in PLOS ONE. Congratulations! Your manuscript is now with our production department. 

Kind regards, 

on behalf of

Gyaneshwer Chaubey 

Academic Editor

PLOS ONE